# Vinculin is critical for the robustness of the epithelial cell sheet paracellular barrier for ions

Satoshi Konishi[1,2,*], Tomoki Yano[1,*], Hiroo Tanaka[1,3,4], Tomoaki Mizuno[1], Hatsuho Kanoh[1,5], Kazuto Tsukita[1,6], Toshinori Namba[7] , Atsushi Tamura[1,3,4], Shigenobu Yonemura[8,9], Shimpei Gotoh[2,10], Hisako Matsumoto[2], Toyohiro Hirai[2], Sachiko Tsukita[1,4]

The paracellular barrier function of tight junctions (TJs) in epithelial cell sheets is robustly maintained against mechanical fluctuations, by molecular mechanisms that are poorly understood. Vinculin is an adaptor of a mechanosensory complex at the adherens junction. Here, we generated vinculin KO Eph4 epithelial cells and analyzed their confluent cell-sheet properties. We found that vinculin is dispensable for the basic TJ structural integrity and the paracellular barrier function for larger solutes. However, vinculin is indispensable for the paracellular barrier function for ions. In addition, TJs stochastically showed dynamically distorted patterns in vinculin KO cell sheets. These KO phenotypes were rescued by transfecting full-length vinculin and by relaxing the actomyosin tension with blebbistatin, a myosin II ATPase activity inhibitor. Our findings indicate that vinculin resists mechanical fluctuations to maintain the TJ paracellular barrier function for ions in epithelial cell sheets.

## Introduction

Every biological compartment is delineated by epithelial cell sheets. In the epithelial cell sheets of vertebrates, the paracellular barrier is structurally and functionally established by continuous belt-like tight junctions (TJs) in the most apical regions of the lateral membranes of the epithelial cells (Farquhar & Palade, 1963; Tsukita et al, 2001; Franke, 2009; Van Itallie & Anderson, 2014; Tsukita et al, 2019). The TJ is constructed on the foundation of the adherens junction (AJ), which also has a belt-like arrangement. The system consisting of the TJ and AJ is collectively referred to as the apical junctional complex (AJC) (Kaye et al, 1966; Ishiuchi & Takeichi, 2011;

Takeichi, 2014). The AJC is critical for various physiological processes and for the paracellular barrier function of the TJ (Buzza et al, 2010; Paschoud et al, 2014). The robustness of the epithelial barrier is maintained under physiological conditions, even when mechanical fluctuations occur at the AJC (Turner et al, 1997; Van Itallie et al, 2009). Although the mechanical roles of the AJ in epithelial cell sheets are well studied in molecular terms (Taguchi et al, 2011; Choi et al, 2016; Bays et al, 2017; Rübsam et al, 2017), its roles in the TJ's paracellular barrier function for maintaining epithelial homeostasis are not well understood.

In the cell–cell AJs, vinculin functions under mechanical fluctuations as an adaptor protein that binds to a mechanically exposed domain of α-catenin, which is associated with β-catenin/cadherin and actin filaments (Yonemura et al, 2010; Yao et al, 2014). Vinculin also plays a role in junctional stability through a tension-induced protective feedback mediated by actomyosin activity (Le Duc et al, 2010; Huveneers et al, 2012; Leerberg et al, 2014). Vinculin down-regulation severely inhibits the TJ formation in mouse teratocarcinoma F9 cells and keratinocytes (Watabe-Uchida et al, 1998; Rübsam et al, 2017). In addition, vinculin is reported to be required for TJ and paracellular barrier maintenance during cytokinesis (Higashi et al, 2016). In kidney-specific conditional KO of vinculin, the integrity of the glomerular filtration barrier formed by podocyte foot processes was reported to be perturbed, although the intercellular junctions in this case did not involve the typical epithelial-type architecture of the TJ (Lausecker et al, 2018). Collectively, vinculin's role in regulating the mechanical fluctuation at AJCs seems to affect the paracellular barrier functions of TJs in the fully confluent condition in vivo.

In this study, we investigated vinculin's role in the TJ's paracellular barrier function in fully confluent epithelial Eph4 cell sheets that have an established paracellular barrier function because of a well-developed belt-like arrangement of AJCs. We

[1]Laboratory of Biological Science, Graduate School of Frontier Biosciences and Graduate School of Medicine, Osaka University, Osaka, Japan   [2]Department of Respiratory Medicine, Graduate School of Medicine, Kyoto University, Kyoto, Japan   [3]Department of Pharmacology, Teikyo University, Tokyo, Japan   [4]Strategic Innovation and Research Center, Teikyo University, Tokyo, Japan   [5]Graduate School of Biostudies, Kyoto University, Kyoto, Japan   [6]Department of Neurology, Graduate School of Medicine, Kyoto University, Kyoto, Japan   [7]Graduate School of Arts and Sciences, Tokyo University, Tokyo, Japan   [8]Department of Cell Biology, Tokushima University Graduate School of Medical Science, Tokushima, Japan   [9]Laboratory for Ultrastructural Research, RIKEN Center for Biosystems Dynamics Research, Kobe, Japan   [10]Department of Drug Discovery for Lung Diseases, Graduate School of Medicine, Kyoto University, Kyoto, Japan

Correspondence: atsukita@biosci.med.osaka-u.ac.jp
*Satoshi Konishi and Tomoki Yano contributed equally to this work

established vinculin KO Eph4 cells and revertant (REV) cells expressing GFP-fused full-length vinculin and found that the AJCs seemed to form correctly in these cells in the fully confluent condition. The TJs' paracellular barrier function for large solutes in the vinculin KO epithelial cell sheets was almost the same as that in the WT and REV cell sheets. However, in the vinculin KO cell sheets, the paracellular barrier function for ions was almost completely lost and parts of the linearly arranged TJs and AJs, unified as AJCs, were stochastically distorted. Our findings indicated that AJ vinculin fine-tunes mechanical fluctuations by actomyosin at the AJC to robustly maintain the TJ's paracellular barrier function for ions, which is critical for epithelial homeostasis.

# Results

### The TJ's paracellular barrier function for large solutes is mostly maintained, but that for ions is lost in vinculin KO epithelial cell sheets

Since the formation of TJs is reported to be affected by vinculin KO in mouse teratocarcinoma F9 cells (Watabe-Uchida et al, 1998), we examined vinculin's role in the TJ's paracellular barrier function by generating vinculin KO Eph4 cells (mammary epithelial cells) using the CRISPR-Cas9 system (Fig S1A). We also generated REV cells by transfecting the KO cells with GFP-tagged full-length vinculin (Fig S1A). We then examined the paracellular barrier function of the TJs in the fully confluent epithelial cell sheet condition. Paracellular tracer flux assays revealed that in WT epithelial cell sheets, the paracellular permeability for large solutes such as fluorescein (0.4 kD) and FITC-dextran (FD-20; 20 kD) was low (Fig 1A). Because low paracellular permeability of an epithelial cell sheet indicates a high level of paracellular barrier function, these results suggested that the WT epithelial Eph4 cell sheets had a well-developed paracellular barrier function for large solutes. We also found that the paracellular permeability of the vinculin KO epithelial cell sheets for large solutes was almost the same as those in the WT and REV epithelial cell sheets, although they were slightly but significantly increased (Fig 1A). In contrast, ZO-1/ZO-2 double-KO (DKO) epithelial cell sheets, which lack TJs, showed a very high permeability for large solutes (Fig 1A). These findings indicated that the paracellular barrier function for large solutes was largely maintained in the vinculin KO epithelial cell sheets in the fully confluent condition.

Next, we examined the paracellular barrier function for ions in WT, vinculin KO, and REV epithelial cell sheets. In the WT epithelial cell sheets, the barrier was very strong, as shown by the high transepithelial electrical resistance (TER) value of 5,500 ~ 6,500 $\Omega \cdot cm^2$, in our culture conditions at full confluency (Fig 1B). However, in the vinculin KO epithelial cell sheets by six different KO clones generated by three different guide RNAs (Fig S1B), the paracellular barrier function for ions was extremely reduced, with a low TER value of 400–500 $\Omega \cdot cm^2$, which was very close in value to that of the TJ-less ZO-1/ZO-2 DKO epithelial cell sheets, with a TER value of 50–150 $\Omega \cdot cm^2$ (Figs 1B and S1B). However, the TER values of vinculin KO epithelial cell sheets were all slightly elevated depending on

time-course, although the TER values of the TJ-less ZO-1/ZO-2 DKO epithelial cell sheets were not. Notably, in the REV epithelial cell sheets, the TER values were recovered to the same level as that in the WT epithelial cell sheets (Fig 1B). These findings indicate that vinculin is dispensable for the TJ paracellular barrier function for large solutes but is indispensable for the paracellular barrier function for ions.

### The continuous belt-like linear TJ pattern is stochastically distorted in vinculin KO epithelial cell sheets

To investigate the mechanism by which the loss of vinculin causes defects in the TJ paracellular barrier function for ions in epithelial cell sheets, we examined the distribution patterns of TJ proteins in WT, vinculin KO, and REV epithelial cell sheets at full confluency. In the WT epithelial cell sheets, TJ proteins such as ZO-1, occludin, and claudin-3 showed a continuous belt-like linear junctional pattern of immunofluorescence (Fig 1C–E). In contrast, in vinculin KO epithelial cell sheets, the immunofluorescence signals for these TJ proteins showed a continuous linear junctional pattern that was stochastically distorted and appeared expanded (Fig 1C–E). This observation suggested that vinculin plays a role in maintaining the continuous linear distribution pattern of TJ proteins, which is required for the TJ paracellular barrier function for ions. At TJs, adhesion molecules called claudins polymerize to form TJ strands, which interact with the TJ strands on adjacent cells to form the paracellular barrier. Freeze-fracture electron microscopy revealed that the TJ strands were almost similarly formed in the WT and vinculin KO epithelial cell sheets (Fig S1C). We also carried out a quantitative analysis of TJ strand according to the previous article (Coyne et al, 2002). There were not significant differences of junctional depth and strand number between WT and vinculin KO Eph4 cells (Fig S1D and E). Consistent with the result that the paracellular barrier function for large solutes, but not for ions, was largely maintained in the vinculin KO epithelial cell sheets (Fig 1A and B), we ascertained that the stochastically distorted linear TJ patterns in vinculin KO cells were at least related to the loss of the TJ paracellular barrier function for ions in the limiting areas. Based on these observations, the findings suggest that vinculin is dispensable for the basic structural integrity of the TJ.

The TJ and AJ are generally integrated in the AJC system to implement the paracellular barrier functions of TJs in epithelial cell sheets (Buzza et al, 2010; Ishiuchi & Takeichi, 2011; Paschoud et al, 2014; Van Itallie & Anderson, 2014). Similar to the TJ proteins, a continuous belt-like linear junctional pattern of AJ proteins, such as E-cadherin, was maintained at all the cell–cell adhesion regions in the WT epithelial cell sheets but was stochastically distorted in the vinculin KO epithelial cell sheets, at the same sites where the linear patterns of TJ proteins were distorted (Fig 2A). The vinculin KO-associated distorted patterns of TJ proteins, such as ZO-1, and of AJ proteins, such as E-cadherin, were restored to the tightly linear patterns in the REV epithelial cell sheets (Fig S2B). Notably, we did not detect any differences in the expression levels of the TJ and AJ proteins (ZO-1, claudins, and E-cadherin) other than vinculin between the WT, vinculin KO, and REV epithelial cell sheets (Fig S1A). In this respect, there is no elevation of the expression of claudin-2 and -15 in mRNA and immunofluorescence (data not shown), which

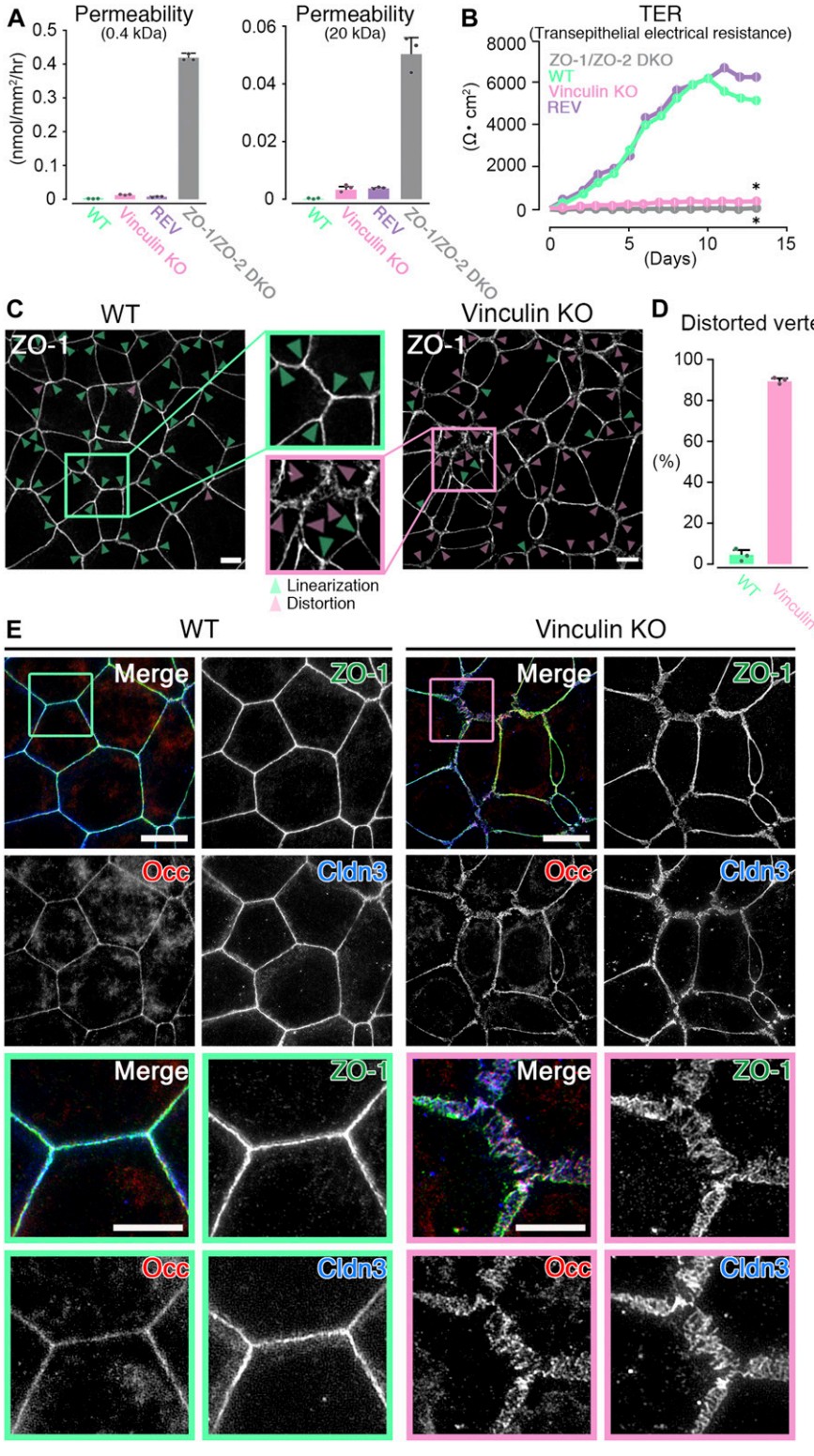

**Figure 1.  The paracellular barrier function for large solutes is largely maintained, but that for ions is lost in vinculin KO epithelial cell sheets.**
**(A)** Paracellular barrier function for large solutes, shown by paracellular tracer flux assays using 0.4-kD fluorescein (left) and 20-kD FD-20 (right), in WT, vinculin KO, REV, and ZO-1/ZO-2 DKO Eph4 epithelial cell sheets. Under the condition of full confluency (14 d of culture on a filter), the paracellular barrier function for fluorescein and FD-20 was largely maintained in vinculin KO cell sheets compared with WT and REV epithelial cell sheets. In contrast, in ZO-1/ZO-2 DKO cell sheets, which lacked TJs, the paracellular barrier function for fluorescein and FD-20 was severely defective. Results from three independent experiments are shown as the mean ± SEM (n = 3/ group). **(B)** Paracellular barrier function for ions, shown by the TER of WT, vinculin KO, REV, and ZO-1/ZO-2 DKO Eph4 epithelial cell sheets. Under the condition of full confluency (13 d of culture on a filter), in the WT and REV cell sheets, the paracellular barrier function for ions was very high (TER of 5,500 ~ 6,500 $\Omega \cdot cm^2$). In contrast, in the vinculin KO cell sheets, the paracellular barrier function for ions was almost completely lost, with a TER (500–600 $\Omega \cdot cm^2$) almost the same as that of the TJ-less ZO-1/ZO-2 DKO epithelial cell sheets. Results from three independent experiments are shown as the mean ± SEM (n = 3/ group). *P*-values were calculated using a two-tailed independent *t* test, and *P* < 0.05 was considered significant. **P* < 0.05. **(C)** Confocal super-resolution immunofluorescence images for ZO-1, a TJ protein, in WT (left) and vinculin KO (right) Eph4 epithelial cell sheets. In WT cell sheets, the ZO-1 signals showed a continuous, belt-like linear junctional pattern (green arrowheads: linear ZO-1 distribution). In contrast, in vinculin KO epithelial cell sheets, the continuous linear junctional pattern of ZO-1 was stochastically distorted to become expanded in the tricellular point (magenta arrowheads: distorted ZO-1 distribution). Scale bars: 10 *μ*m. **(D)** The ratio of distorted vertices to all the tricellular vertices of each cell sheets was calculated by representative images from three independent experiments and described. **(E)** Confocal super-resolution immunofluorescence images for TJ proteins ZO-1, occludin (Occ), and claudin-3 (Cldn3) in WT (left) and vinculin KO (right) Eph4 epithelial cell sheets. In WT epithelial cell sheets, ZO-1 (green), occludin (red), and claudin-3 (blue) showed a continuous belt-like linear junctional pattern of immunofluorescence. In contrast, in vinculin KO epithelial cell sheets, the continuous linear junctional patterns of ZO-1 (green), occludin (red), and claudin-3 (blue) were stochastically distorted to then become expanded. Lower images are magnifications of the boxed regions in the upper images. Representative images from three independent experiments are shown. Scale bars: 10 *μ*m.

denied the possibility that decreased TER in vinculin KO cells was because of the elevation of the expression of the channel-type claudins. Taken all together, it was most likely that the stochastically distorted patterns of TJ and AJ proteins because of the loss of vinculin were related to the defects in the TJs' paracellular barrier function. These observations also indicated that the TJ and AJ were consistently integrated in the AJC, even in the vinculin KO cell sheets.

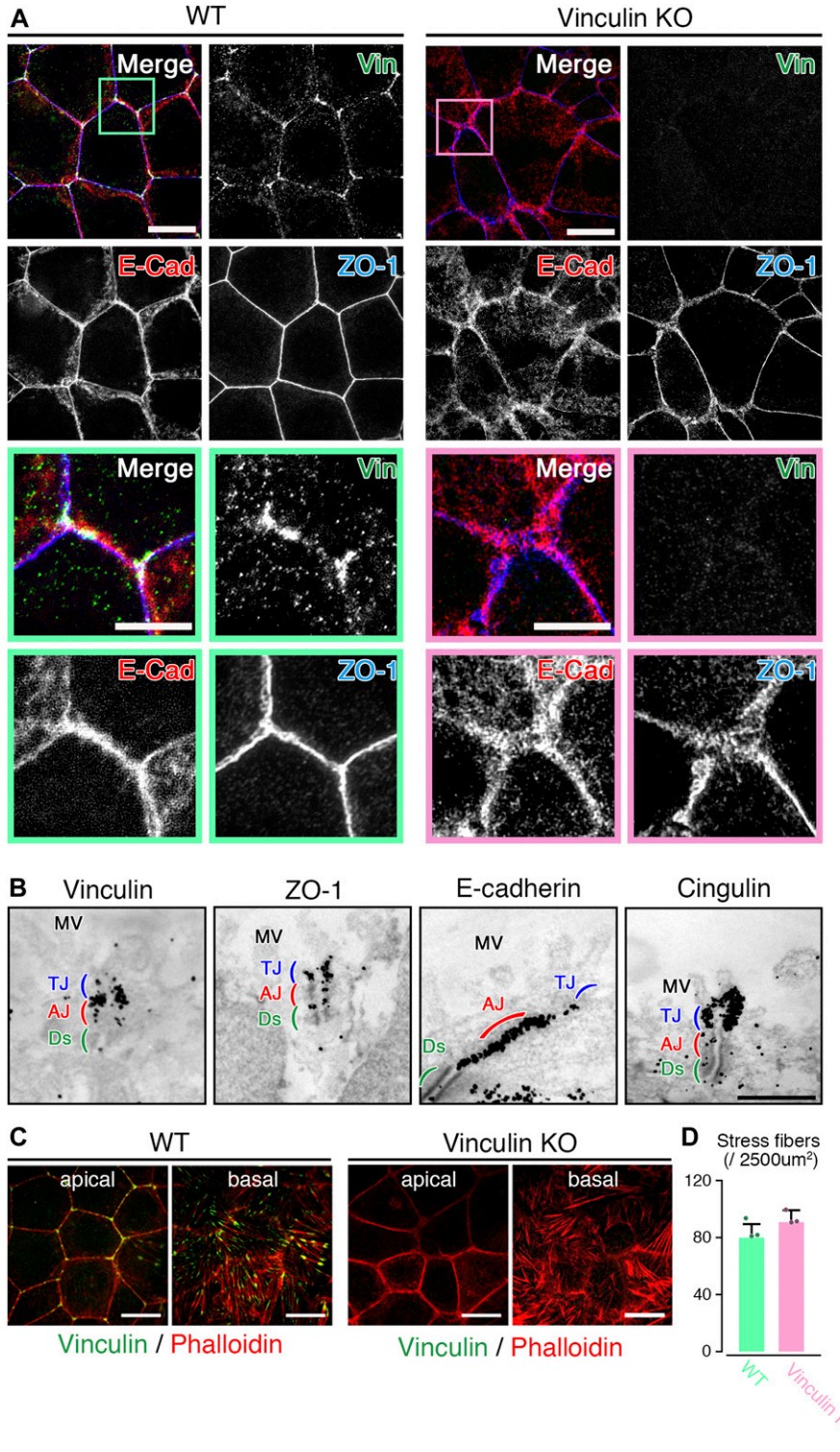

**Figure 2. The linear distribution patterns of AJ and TJ proteins are stochastically distorted to become expanded in vinculin KO epithelial cell sheets.**
**(A)** Confocal super-resolution immunofluorescence images of an AJ protein E-cadherin (E-Cad), a TJ protein ZO-1, and vinculin (Vin) in WT (left) and vinculin KO (right) Eph4 epithelial cell sheets. In WT epithelial cell sheets, vinculin (green) was co-localized with E-cadherin (red) and ZO-1 (blue) at the most apical parts of the lateral membranes and was particularly concentrated at tricellular junctions. In vinculin KO epithelial cell sheets, the vinculin was completely lost and the distribution pattern of E-cadherin (red) was stochastically distorted to then become expanded. Lower images are magnified images of the boxed regions in the upper images. Representative images from three independent experiments are shown. Scale bars: 10 μm. **(B)** Immunoelectron microscopy images of AJ and TJ proteins vinculin, ZO-1, E-cadherin, and cingulin in WT Eph4 epithelial cell sheets. Immunogold signals for vinculin were predominantly observed in the AJ (red) with a very few signals in the TJ (blue). The AJ protein E-cadherin and TJ protein cingulin were predominantly localized to the AJ and TJ, respectively. Notably, immunogold signals for ZO-1 were observed in both the TJ and the AJ. Representative images from three independent experiments are shown. MV, microvilli; Ds, desmosome. Scale bar: 500 nm. **(C)** Confocal super-resolution immunofluorescence images of phalloidin, an F-actin marker, and vinculin in WT (left) and vinculin KO (right) Eph4 epithelial cell sheets. In WT epithelial cell sheets, vinculin (green) accumulated especially at tricellular junctions in the apical side of the cells and was also localized to the edge of stress fibers, indicated by phalloidin (red), at the basal side of the cells. Scale bars: 10 μm. **(D)** The number of stress fiber in the z projection image of the basal side of the cells recorded by super-resolution confocal microscopy was measured. Representative images from three independent experiments were used. There was no significant difference in the stress fiber formation between WT and vinculin KO epithelial cell sheets.

## Vinculin is mainly localized to the AJ and has a regulatory role in the TJs' paracellular barrier function on the basis of AJC

Our findings reveal an apparent role of vinculin in the TJ's paracellular barrier function for ions (Fig 1B). However, vinculin is generally accepted to be highly localized to cell–cell AJs (Geiger et al, 1981). Therefore, we examined the localization of vinculin in the context of the AJC at cell–cell junctions in fully confluent epithelial cell sheets. Confocal super-resolution microscopy revealed that in the WT epithelial cell sheets, vinculin was co-localized with

AJC proteins such as ZO-1 and E-cadherin at the most apical parts of the lateral membranes and was particularly concentrated at tricellular junctions (Fig 2A).

Next, we examined the spatial distribution of vinculin in AJCs in detail by immunoelectron microscopy, in which the TJ and AJ can be clearly identified by their different morphological features. In the WT epithelial cell sheets, immunogold signals for vinculin were predominantly observed in the AJ, with very few signals in the TJ (Fig 2B). Immunoelectron microscopy for other junctional proteins in the WT epithelial cell sheets revealed that AJ proteins (E-cadherin and β-catenin) and TJ proteins (cingulin and occludin) were predominantly localized to the AJ and TJ, respectively (Figs 2B and S2A).

Notably, the immunogold signals for ZO-1 were observed in both the TJ and the AJ in the WT epithelial cell sheets (Fig 2B). In this respect, the result of the immunoprecipitation assays using exogenously transfected HEK cells could indicate that ZO-1 binds to the D2-4 domains of vinculin (Fig S2C). Consistently, it was previously shown that vinculin binds to ZO-1 in the AJ of cardiomyocytes (Zemljic-Harpf et al, 2014). When we also compared the distribution of ZO-1 of the immunogold signals between WT and vinculin KO Eph4 epithelial cells, there was no significant difference in the distribution of ZO-1 signals, indicating that vinculin does not change the localization of ZO-1 in the AJC (Fig S2D). In addition, because vinculin is known to be localized not only in cell–cell junctions but also in cell–substrate focal adhesions (Geiger, 1979; Burridge & Feramisco, 1980), we compared the characteristics of the focal adhesions in the WT versus vinculin KO epithelial cell sheets and observed no significant differences in their stress fibers (Fig 2C and D). These findings supported the idea that vinculin has a role in the TJs' paracellular barrier function for ions, on the basis that the TJ and AJ are topologically and functionally integrated into the AJC as a unified system.

### The transition between linear and distorted TJ patterns in vinculin KO epithelial cell sheets is dynamic

The finding that the distorted TJ patterns appeared stochastically in the vinculin KO epithelial cell sheets (Fig 1C and D) prompted us to examine the dynamics of the TJ patterns using live cell imaging. For this analysis, we generated WT and vinculin KO epithelial cells that stably expressed Venus-tagged occludin to mark the TJs. Live cell imaging revealed that the Venus–occludin-expressing vinculin KO epithelial cells showed much more movement than the Venus–occludin-expressing WT epithelial cells in full confluency (Fig 3A and B and Videos 1 and 2). In particular, we noted stochastic transitions between the tightly linear and expanded distorted pattern of Venus–occludin in the vinculin KO epithelial cell sheets (Fig 3B and Video 2). To analyze the difference in the dynamics between WT and vinculin KO epithelial cells quantitatively, we selected epithelial cells with similar areas and measured the time-averaged mean squared displacement (tMSD) of the vertices of the epithelial cells (Figs 3C and S3A). This analysis revealed that the tMSD of the vertices of vinculin KO epithelial cells was larger than that of WT epithelial cells (Fig 3D), confirming that vinculin

KO cells showed much more movement of cell–cell border than the WT cells.

These findings indicated that vinculin at AJs is required to resist a tendency for TJs to become distorted and expanded, thereby maintaining their continuous tightly linear arrangement, which is critical for the TJ's paracellular barrier function for ions.

### Defects in the linear TJ distribution in vinculin KO cells are reversed by tension-relaxing blebbistatin treatment

Accumulating evidence indicates that vinculin binds to a mechanically stretched α-catenin to resist mechanical fluctuation at AJs (Yonemura et al, 2010). Therefore, we speculated that mechanical fluctuation was involved in the distorted and expanded TJ pattern in the vinculin KO epithelial cell sheets. To investigate this possibility, we visualized mechanical fluctuations in the WT and vinculin KO epithelial cell sheets using the anti-α-catenin monoclonal antibody (α18), which binds to mechanically stretched α-catenin (Yonemura et al, 2010). In the WT epithelial cell sheets, immunofluorescent signals for the α18 staining normalized by α-catenin staining showed a continuously linear pattern with high intensity at tricellular junctions, similar to the pattern for vinculin (Fig 4A). In the vinculin KO epithelial cell sheets, the α18 staining normalized by α-catenin staining was strongly detected in the distorted TJ regions and were significantly increased compared with those in the WT epithelial cell sheets (Fig 4A). The α18 staining was markedly more widely distributed in the WT cells than in the vinculin KO cells, and the α18 staining was more concentrated at tricellular regions in vinculin KO cells than in WT cells (Fig 4A and B). These observations indicated that in the vinculin KO epithelial cell sheets, mechanical fluctuation around the tricellular junctions was particularly greater than that in WT and apparently distorted the continuous, linear TJ pattern.

We next asked whether a reduction in mechanical fluctuation could prevent the transition from the linear to the distorted TJ pattern in the vinculin KO epithelial cell sheets. To address this question, we treated the WT and vinculin KO epithelial cell sheets with blebbistatin, which relaxes actomyosin-induced tension. In WT cells treated with blebbistatin, the fluorescent signals for both vinculin and α-catenin/α18 staining were significantly decreased (Fig 4C). In the vinculin KO cells, blebbistatin also caused the staining of the α18 to significantly decrease. These findings indicated that blebbistatin treatment reduced the tension at AJCs in WT and vinculin KO cells.

Examination of the AJC patterns revealed that under blebbistatin treatment their distribution appeared to be very similar between the vinculin KO and WT epithelial cell sheets, whereas they were different in the control (DMSO-treated) cell sheets: linear in WT cells and stochastically distorted in vinculin KO cells (Figs 4D and S3B and C). Taken with the live cell imaging results (Videos 1 and 2), our findings suggest that mechanical fluctuation stochastically and transiently induces the distorted and expanded patterns of TJs in vinculin KO cells in normal culture conditions without blebbistatin treatment. These findings strongly suggest that vinculin mechanically fine-tunes the position of AJCs to maintain their linear distribution.

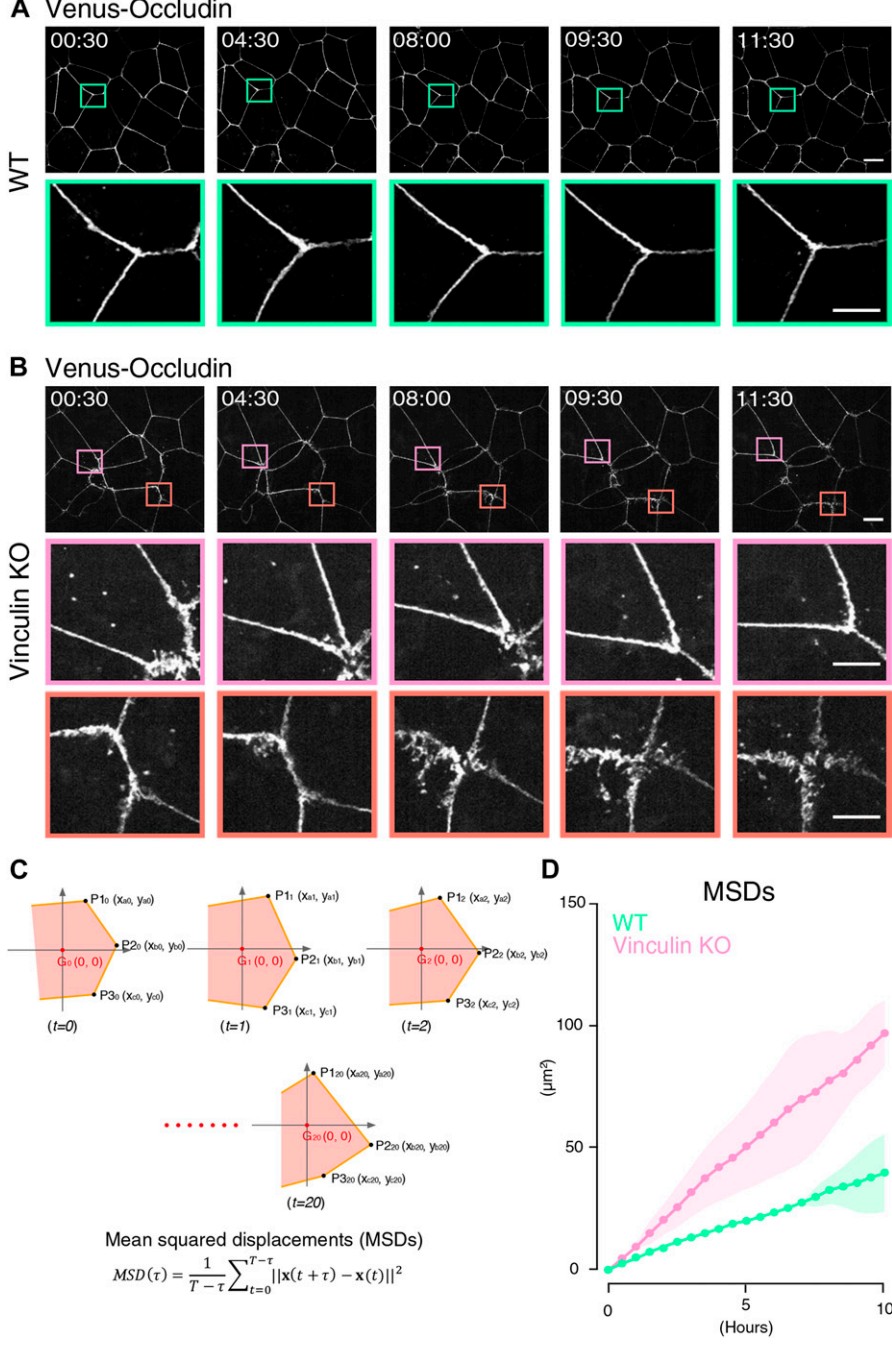

**Figure 3. The transition between linear and distorted TJ patterns in vinculin KO epithelial cell sheets is dynamic.**

**(A)** Live cell imaging of WT Eph4 epithelial cells stably expressing Venus-tagged occludin as a marker for TJs (see Video 1). In Venus–occludin-expressing WT epithelial cell sheets, the cell shape and continuous linear distribution pattern of Venus–occludin were maintained. Lower images are magnifications of the boxed regions in the upper images. Representative time-lapse images of Venus–occludin at the indicated time points from three independent experiments are shown. Time is shown in hours: minutes. Scale bars: 10 $\mu$m. **(B)** Live cell imaging of vinculin KO Eph4 epithelial cells stably expressing Venus-tagged occludin as a marker for TJs (see Video 2). In Venus–occludin-expressing vinculin KO epithelial cell sheets, the TJ patterns showed much more movement than the Venus–occludin-expressing WT epithelial cells. Notably, stochastic transitions from the distorted to linear pattern of Venus–occludin (magenta boxes) and from the linear to distorted pattern of Venus–occludin (orange boxes) were observed in Venus–occludin-expressing vinculin KO cell sheets. Lower images are magnifications of the boxed regions in the upper images. Representative time-lapse images of Venus–occludin at the indicated time points from three independent experiments are shown. Time is shown in hours: minutes. Scale bars: 10 $\mu$m. **(C)** Calculation of the tMSD of the vertices of Venus–occludin-expressing WT and vinculin KO Eph4 epithelial cells. **(D)** Quantitative analysis of the tMSD of the vertices of Venus–occludin-expressing WT and vinculin KO Eph4 epithelial cells. We selected epithelial cells with similar areas (Fig S3A). The tMSD of the vertices of Venus–occludin-expressing vinculin KO epithelial cells was larger than that of Venus–occludin-expressing WT epithelial cells (Fig 3C), confirming that the vinculin KO epithelial cells showed much more movement than the WT epithelial cells. Results from three independent experiments are shown as mean ± SEM (n = 3/group).

### Defects in the paracellular barrier function for ions in vinculin KO cells are reversed by tension-relaxing blebbistatin treatment

When we examined the paracellular barrier function in vinculin KO and WT epithelial cell sheets, we found that blebbistatin treatment restored the paracellular barrier function for ions in the vinculin KO epithelial cell sheets (Fig 5A). This finding indicated that vinculin finely tunes the mechanical fluctuation to maintain the continuous linear pattern of the TJs and the TJs' paracellular barrier function for ions. However, the very small decrease in the paracellular barrier function for large solutes in the vinculin KO epithelial cell sheets was not recovered by blebbistatin treatment (Fig 5B), suggesting that the mechanisms are different for the paracellular permeability for solutes and ions.

Finally, based on the previous finding that vinculin's actin-binding domain (ABD) can be replaced with $\alpha$-catenin's ABD in supporting AJ formation (Yonemura et al, 2010), we investigated whether the ABD of vinculin participates in its role in the TJ's paracellular barrier function via AJ formation. We found that a fusion molecule of $\alpha$-catenin (1–848 aa) without an indispensable site for actin binding fused to vinculin's

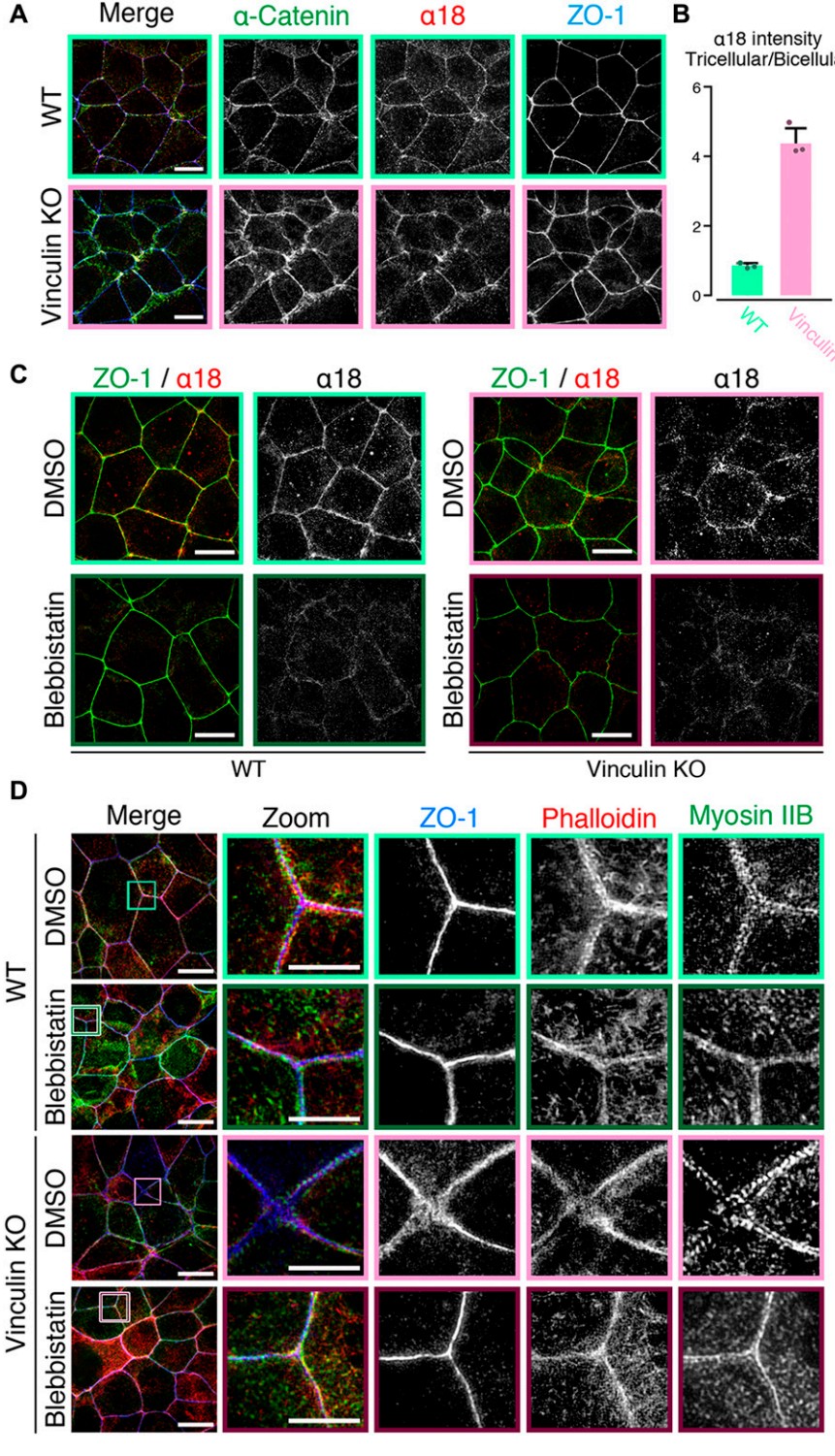

**Figure 4. Mechanical fluctuation induces the distorted TJ patterns in vinculin KO epithelial cell sheets.**

**(A)** Confocal super-resolution immunofluorescence images of α18 antigen, which indicates mechanical fluctuation, in WT (upper) and vinculin KO (lower) Eph4 epithelial cell sheets. The α18-antigen signals were concentrated at tricellular regions, as shown by a TJ protein ZO-1 (red), and DAPI-stained nuclei (blue), in WT and vinculin KO epithelial cell sheets. Notably, in vinculin KO cell sheets, α18-antigen signals were strongly detected at the distorted TJ regions and were significantly increased compared with those in WT epithelial cell sheets. Right images are magnifications of the boxed regions in the left images. Representative images from three independent experiments are shown. Scale bars: 10 μm. **(B)** The signal intensity of immunostained α18 antibody normalized by the signal intensity of immunostained α-catenin antibody was measured at the bicellular points and tricellular points both in WT and in vinculin KO Eph4 cell sheets. The results of the representative images of three independent experiments are shown. The signal intensity of recorded images was measured without adjustments of brightness and contrast. In vinculin KO cell sheets, the signal intensity of α18 normalized by α-catenin was higher in the tricellular points than in the bicellular points. **(C)** Confocal super-resolution immunofluorescence images of Myosin-IIB, an F-actin marker phalloidin, and a TJ protein ZO-1 in WT (upper) and vinculin KO (lower) Eph4 epithelial cell sheets treated with blebbistatin, an actomyosin tension-relaxing reagent, or DMSO. In the vinculin KO cell sheets treated with 100 μM blebbistatin, the expanded and distorted distribution patterns of myosin IIB (green), phalloidin (red), and ZO-1 (blue) were recovered to linear distribution patterns along cell–cell junctions. Right images are magnifications of the boxed regions in the left images. Representative images from three independent experiments are shown. Scale bars: 10 μm. **(D)** Confocal super-resolution immunofluorescence images of α18-antigen, which indicates mechanical fluctuation, in WT (left) and vinculin KO (right) Eph4 epithelial cell sheets. Eph4 epithelial cell sheets treated with blebbistatin (lower) or DMSO (upper). The α18 antigen signals were concentrated at tricellular regions, as shown by a TJ protein ZO-1 (red), in WT and vinculin KO epithelial cell sheets in DMSO control. The α18-antigen signals were significantly decreased both in WT and in vinculin KO cells (lower). Representative images from three independent experiments are shown. Scale bars: 10 μm.

ABD partially restored the paracellular barrier function for ions, as shown by TER, in vinculin KO epithelial cell sheets. The transfection of C-terminal ABD of vinculin to vinculin KO cells and the transfection of α-catenin without indispensable site for actin binding showed no effect in TER in vinculin KO cells (Fig 5C). The finding that a fusion molecule of the C-terminal ABD of vinculin and α-catenin (1–848 aa)

partially restored the paracellular barrier function indicated that vinculin in AJ has effects on TJ function through its ABD.

Taking all these results into consideration, we propose that vinculin at the AJ fine-tunes the mechanical fluctuations by actomyosin at cell junctions to maintain the TJ's robust paracellular barrier function for ions and the linear distribution of

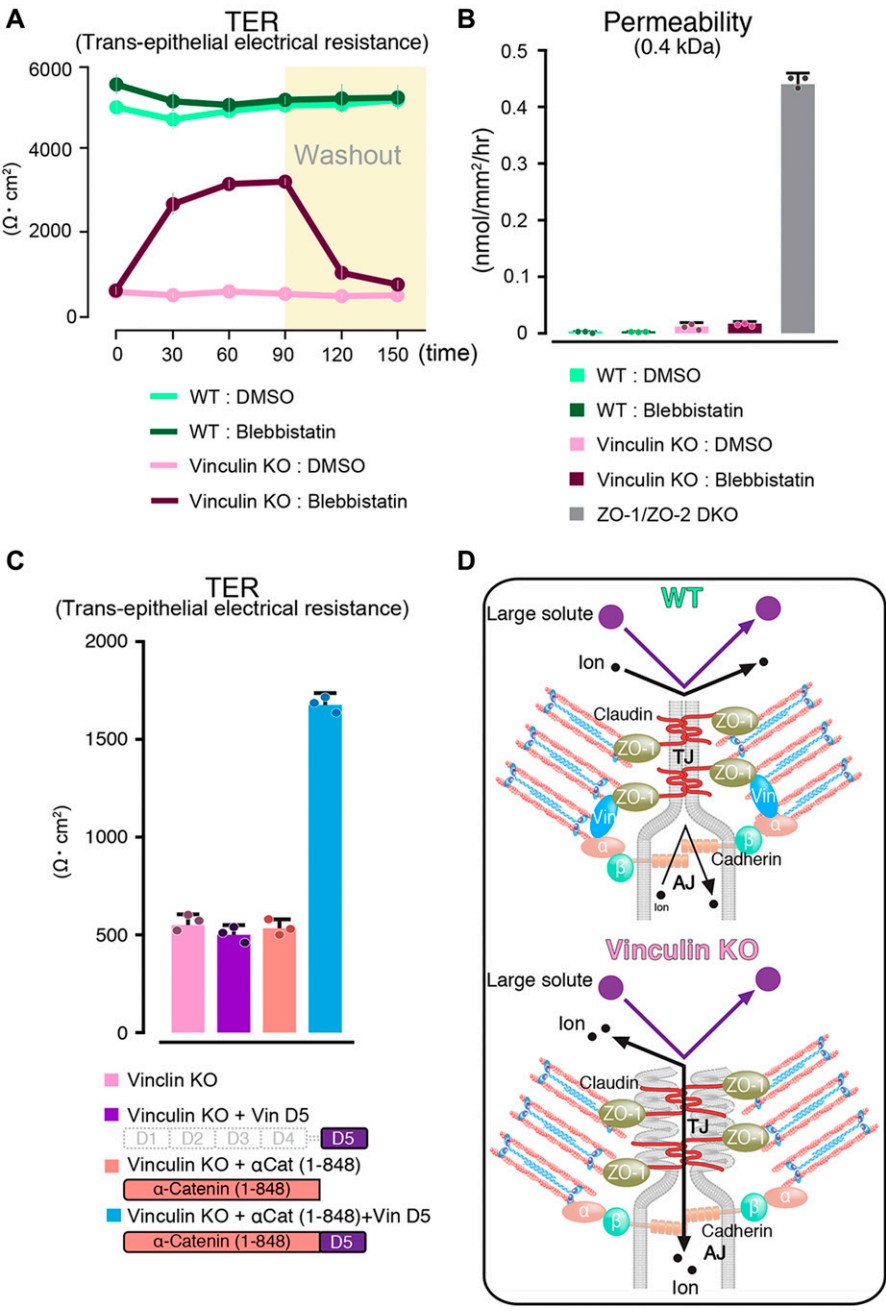

**Figure 5. Mechanical fluctuation induces the loss of the paracellular barrier function for ions in vinculin KO epithelial cell sheets.**
**(A)** The paracellular barrier function for ions, shown by the TER of WT and vinculin KO Eph4 epithelial cell sheets treated with blebbistatin, an actomyosin tension-relaxing reagent, or DMSO. Under the condition of full confluency (13 d of culture on a filter), blebbistatin (100 $\mu$M) but not DMSO significantly rescued the TER reduction in the vinculin KO epithelial cell sheets. Results from three distinct clones are shown as the mean ± SEM (n = 3/group).
**(B)** Paracellular barrier function for large solutes, shown by paracellular tracer flux assays using 0.4-kD fluorescein, in WT and vinculin KO Eph4 epithelial cell sheets treated with blebbistatin, an actomyosin tension-relaxing reagent, or DMSO and in ZO-1/ZO-2 DKO Eph4 epithelial cell sheets. Under the condition of full confluency (13 d of culture on a filter), the slight increase in the paracellular permeability for 0.3-kD fluorescein in the vinculin KO epithelial cell sheets was not restored by blebbistatin (100 $\mu$M). Results from three distinct clones are shown as the mean ± SEM (n = 3/group). **(C)** Scheme of the molecules used to transfect vinculin KO Eph4 cells and the TER measured in the monolayers of filter-grown vinculin KO cells and vinculin KO stable clones after transfection.
**(D)** Schematic images showing the AJC as a unified system that maintains the TJ's paracellular barrier function for ions. Vinculin at the AJ fine-tunes the mechanical fluctuations to maintain the robust, linear distribution of the AJC, which is indispensable for the TJ's paracellular barrier function for ions. In this mechanism, vinculin at the AJ may be connected with the TJ via ZO-1 to fine-tune the mechanical fluctuation at the AJC. $\alpha$, $\alpha$-catenin; $\beta$, $\beta$-catenin.

AJCs (Fig 5D). In this mechanism, vinculin at the AJ may link with the TJ via ZO-1 to regulate the TJ's paracellular barrier function for ions because it was localized to both the AJ and the TJ and has been shown to bind to TJ claudins, AJ $\alpha$-catenin, and AJ vinculin (Fig S2C).

## Discussion

Various cell contexts, including contraction, motility, signal transduction, and other circumstances, are known to affect or regulate the TJ paracellular barrier function (Turner, 2000; Zhang et al, 2006; González-Mariscal et al, 2008; Escudero-Esparza et al, 2012; Shigetomi & Ikenouchi, 2017; Shigetomi et al, 2018; Yano et al, 2018). The actomyosin of epithelial cells is connected to TJ- and AJ-associated proteins (Hartsock & Nelson, 2008), and the mechanical properties of the AJC have recently been gaining attention (Taguchi et al, 2011; Choi et al, 2016; Bays et al, 2017; Cartagena-Rivera et al, 2017; Rübsam et al, 2017; Spadaro et al, 2017; Kale et al, 2018). However, little is known about how paracellular barrier functions are robustly maintained against the mechanical fluctuations.

In the present study, we discovered that the paracellular barrier for ions was disrupted in vinculin KO Eph4 cells and attributed this loss to a defect in the AJCs, the cell–cell junctions of TJs and AJs rather than in focal adhesions (Fig 2). We further found that the AJC defects in vinculin KO cells varied with time and location in the cell sheet and were rescued by the myosin II ATPase activity inhibitor, blebbistatin (Fig 4). Considering the previous findings that the epithelial paracellular barrier function is regulated by myosin light chain kinase via the ABD of ZO-1 (Tamada et al, 2007; Yu et al, 2010), our analyses indicated that vinculin helps to maintain the continuous belt-like TJs that create the paracellular barrier for ions by fine-tuning the actomyosin-dependent mechanical fluctuations.

In the current study, our analyses of vinculin KO cell sheets revealed that vinculin had distinct roles in regulating the TJ paracellular barrier for ions versus solutes. The paracellular barrier function of TJs also involves paracellular permeability, of which there are two types: "pore" pathways, in which solutes or ions pass through a paracellular channel formed in the TJ strands, and "leak" pathways, in which they presumably pass through breaks in the TJ strands (Sasaki et al, 2003; Suzuki et al, 2015; Weber & Turner, 2017; Zhao et al, 2018). Our findings imply that in vinculin KO cells, the typical "pore" pathways for ions and solutes, which usually involve specific claudins, do not form correctly. However, the "leak" pathways, which may be generated by mechanical fluctuations by actomyosin, may still be functional in vinculin KO cells.

Our present findings suggest that a vinculin-based system is involved in maintaining the robust paracellular barrier function of epithelial cell sheets under mechanical fluctuations by actomyosin. The TJ paracellular barrier function is regulated by the AJ through the coordination of TJ and AJ components within the AJC. Further studies of the epithelial barrier system will improve our understanding of the mechano-related regulations responsible for the robustness of the physiological epithelial paracellular barrier functions.

# Materials and Methods

### Antibodies and reagents

The primary antibodies used in this study included mouse anti-vinculin (VIN-11-5) mAb (Sigma-Aldrich), rabbit anti-claudin-3 pAb (Invitrogen), rabbit anti-ZO-1 (Invitrogen), mouse anti-$\beta$-catenin mAb (BD), rabbit anti-myosin heavy chain IIB pAb (Covance), mouse anti-$\beta$-actin mAb (Sigma-Aldrich), rat anti-HA (3F10) mAb (Roche), mouse anti-GFP mAb (Invitrogen), and rabbit anti-RFP pAb (Genetex). The rat anti-occludin (MOC37) mAb, mouse anti-ZO-1 (T-8) mAb, and rat anti-ZO-1 mAb were generated in our laboratory (Itoh et al, 1991; Saitou et al, 1998; Kitajiri et al, 2004). The rat anti-E-cadherin (ECCD2) mAb was provided by Dr M Takeichi (Riken, BDR) and rat anti-$\alpha$-18 mAb was gifted by Dr A Nagafuchi (Nara Medical University). Secondary antibodies labeled with Alexa Fluor 488, 568, and 647 (Invitrogen), Cy3-labeled secondary antibody (Jackson Immunoresearch), and rhodamine-conjugated phalloidin (Invitrogen) were purchased.

### Plasmid construction

To construct the vinculin–EGFP and occludin–Venus plasmids, full-length vinculin and occludin were amplified from mouse liver cDNA library by PCR, respectively. Vinculin–EGFP and occludin–Venus were constructed by ligating PCR-amplified samples into pEGFP-N3 vector and pCAGGS-Venus vector, respectively.

### Cell culture and transfection

Murine mammary epithelial cells (Eph4 cells) and HEK293 cells were cultured in DMEM supplemented with 10% FCS. Transfection was performed using Lipofectamine Plus or Lipofectamine 2000 (Thermo Fisher Scientific) following the manufacturer's instructions. To establish stable transfectants, the transfected cells were selected by incubation in medium containing 500 $\mu$g/ml G418 (Nacalai Tesque) and clones derived from single cells were picked up.

### Generation of the vinculin KO cell line using KO constructs

We used the CRISPR-Cas9 vector, pX330 plasmid (Addgene). We first searched for protospacer adjacent motif sequences in the murine vinculin exon sequences (National Center for Biotechnology Information Reference Sequence: NM_009502.4, NP_033528.3). The target sequences for the guide RNA were 18–40 for oligo 1 (Vinculin KO-1,2), 24–46 for oligo 2 (Vinculin KO-3,4), and 48–70 for oligo 3 (Vinculin KO-5,6). Each of these oligonucleotides was annealed and cloned into the pX330 vector, which was cut with BbsI. Each KO construct was transfected into Eph4 cells and subcloned in 96-well. Correct cloning and sequences were confirmed by capillary electrophoresis sequencing of the constructs, and the disruption of the vinculin protein expression was also confirmed by immunofluorescence experiments and Western blotting. The revertant cells of Vinculin KO cells were generated by the G418 selection of vinculin KO Eph4 cells transfected with vinculin-expression plasmids.

### Immunostaining and confocal laser scanning microscopy

The cells were fixed in 1% formalin for 8 min at RT followed by treatment with 0.1% Triton X-100 in PBS. After blocking for 10 min in blocking buffer (PBS containing 1% BSA), the cells were incubated with primary antibodies in blocking buffer for 1 h at RT or overnight at 4°C. The cells were then washed three times with PBS, followed by incubation with fluorochrome-conjugated secondary antibodies for 1 h at RT. The cells were then mounted in fluorescence mounting medium (Dako). The immunofluorescently stained samples were imaged with a Olympus spinning disk super-resolution microscope (SD-OSR; Olympus) equipped with a silicon oil-immersion objective lens (UPlanSApo 60× Sil, NA 1.3; Olympus) with a 1.6× conversion lens, a sCMOS camera (ORCA-Flash 4.0 v2; Hamamatsu Photonics), appropriate filter sets for DAPI/FITC/TRITC and a motorized scanning deck. Image acquisition was set to a $z$-series of multiple planes with a 0.25-$\mu$m distance. Immunofluorescently stained samples were imaged using a silicon oil-immersion objective lens (UPlanSApo 60×Sil, NA 1.3; Olympus). All of the hardware was

controlled with MetaMorph (Molecular Devices). The resulting 3D images were converted into 2D images by maximum intensity projection in the *z*-direction before the image analysis. The images were analyzed with MetaMorph (Molecular Devices).

## Measurement of TER

For the TER measurements, Eph4 cells were plated on Transwell permeable supports (1 × 10⁵ cells on each 12 mm-diameter poly-carbonate support with a 0.4 *μm* pore size) (Costar), and grown at 37°C in a 95% air, 5% $CO_2$ environment. The culture medium was exchanged every day. The TER was directly measured in the culture medium every day using a Millicell-ERS Epithelial Volt-Ohm Meter (EMD Millipore).

## Measurement of the transepithelial flux of fluorescent tracers

Eph4 cells were cultured on Transwell permeable supports for 14 d as described above. To examine the transepithelial flux of fluorescent tracers, the cell layer on the Transwell permeable support was placed into an Ussing chamber with a 5 μm-diameter pore. The basal side of the chamber was filled with 3 ml of Solution A (140 mM NaCl, 5 mM glucose, 5 mM KCl, 1 mM $MgCl_2$, 1 mM $CaCl_2$ and 10 mM HEPES-NaOH [pH 7.4]). The apical side of the chamber was filled with 3 ml of Solution A containing the fluorescent tracer fluorescein (Sigma-Aldrich) or FD-4 (final concentration, 0.5 mM) (Sigma-Aldrich). The temperature was maintained at 37°C and 100% $O_2$ was bubbled through the solution continuously. We collected 200 *μ*l of the solution in the basal side of the chamber every 20 min from 0 to 80 min. The fluorescence of the collected sample was determined at an excitation wavelength of 495 nm by a microplate reader SH-9000 (Hitachi) and then the transepithelial flux for the fluorescent tracer was calculated.

## Live cell imaging

The movement of occludin–Venus Eph4 cells was recorded using a SD-OSR microscope (Olympus; Hayashi & Okada, 2015) equipped with a silicon oil-immersion objective lens (UPlanSApo 60×Sil, NA 1.3; Olympus), a sCMOS camera (ORCA-Flash 4.0 v2; Hamamatsu Photonics), a laser with an excitation wavelength of 488 nm, a motorized scanning deck, and an incubation chamber (37°C; 5% $CO_2$; 85% humidity; Tokai Hit). The recording was performed by multistage acquisitions with 10–30 fields of view, each field covering ~10 cells. The time intervals were 30 min for long-term recording. Image acquisition was set to a *z*-series of 8–12 planes with a 1.0-*μm* distance. The resulting 3D images were converted into 2D images by maximum intensity projection in the *z*-direction before the image analysis.

## Analysis of mean squared displacement

The live cell imaging data of the occludin–Venus WT and vinculin KO Eph4 cells that was recorded every 30 min were analyzed. To measure the cell-shape fluctuation, we first measured the areas of

cells using image J plugin software (National Institutes of Health) and selected cells whose area was nearly the same and then tracked all the vertices in the selected cells. From this time series, we calculated the tMSDs for each vertex, and the ensemble average of these tMSDs was regarded as the cell fluctuation. tMSDs for each vertex was defined as:

$$MSD(\tau) = \frac{1}{T-\tau}\sum_{t=0}^{T-\tau}\|\mathbf{x}(t+\tau) - \mathbf{x}(t)\|^2,$$

where *MSD* is the mean squared displacement, $\mathbf{x}(t)$ is the Euclidean vector at time *t*, *τ* is the lag time, and $\|...\|$ is the Euclidean norm. Cells with the same average area (*A*) were selected (900 ≤ A ≤ 1,400 [$\mu m^2$]).

## Blebbistatin treatment

For immunostaining experiments, 100 *μ*M blebbistatin (Wako) was applied to Eph4 cells for 2 h and then the cells were analyzed. For TER and paracellular flux experiments, 100 *μ*M blebbistatin was applied to Eph4 cells for 2 h, then replaced with DMEM supplemented with 10% FCS after washing the cells three times.

## Immunoelectron microscopy

Cells were fixed in 1% formalin for 8 min at RT followed by treatment with 5% saponin in HEPES-buffered saline for 10 min. After blocking for 5 min in blocking buffer containing 5% saponin in Block Ace (DS Pharma Biomedical Co., Ltd), the cells were incubated with primary antibodies in the blocking buffer for 2 h at 37°C, followed by an incubation with secondary antibodies diluted in blocking buffer for 2 h at 37°C. The cells were then fixed in a solution containing 2% paraformaldehyde, 2.5% glutaraldehyde, 0.5% tannic acid, and 0.1 M HEPES buffer for 1 h at RT and washed with 0.1 M HEPES buffer (pH 7.5) and then with 1 M HEPES buffer (pH 5.8). The cells were then mounted in reagent from the HQ Silver Enhancement Kit (Nanoprobes). The lipid of the cells was then fixed with 1% $OsO_4$ in 0.1 M HEPES buffer (pH 7.5) for 2 h on ice. The samples were then dehydrated and embedded in Poly/Bed 812 (Polysciences).

## Freeze-fracture electron microscopy

Confluent cell sheets were fixed with 2.5% glutaraldehyde in 100 mM sodium phosphate buffer (pH 7.5) at 4°C overnight. The samples were then rinsed with phosphate buffer, mixed with 30% glycerol prepared in phosphate buffer, and frozen in liquid propane. The frozen samples were fractured at −110°C and coated with carbon at a 45° angle, followed by platinum shadowing at a 90° angle in an EM19500-NFSDT (JEOL). Cells were removed from the replica samples by immersion in household bleach. The replica samples were mounted on grids, and images were acquired with a JEM1400 plus electron microscope (JEOL) at an acceleration voltage of 80 kV. When we quantify junctional depth of TJ strands, TJ strands were transected by lines perpendicular to strands. Junctional depths were measured from the upper side of strand to lower side of strand in electron microscope images. Strand number was counted from the upper side of strand to the lower side of strand.

### Immunoprecipitation

HEK293 cells were transfected with expression vectors. The cell lysates were then incubated with protein A-Sepharose bound to anti-vinculin, anti-ZO-1, or anti-HA antibodies. The immune complexes were fully washed and then resuspended in 30 $\mu$l of SDS sample buffer, and 5 and 20 $\mu$l aliquots of each sample were analyzed by Western blot.

### SDS–PAGE and Western blot assay

To prepare the total cell lysates for immunoblotting, Eph4 or HEK293 cells, or Venus-tagged vinculin and ZO-1–expressing transfected Eph4 cells were lysed with SDS–PAGE sample buffer, sonicated, and boiled. The protein samples were separated by SDS–PAGE, transferred onto a nitrocellulose or a polyvinylidene difluoride membrane, and blotted with the appropriate antibodies. Protein bands were detected by Western Lightning Plus-ECL (PerkinElmer). Signals were acquired using an ImageQuant LAS 4000 instrument (GE Healthcare). For quantification of the Western blotting signals, the bands along with a loading control in the same immunoblot membrane were subjected to densitometry and analyzed using ImageJ software (National Institutes of Health).

### Statistical analysis

Data are presented as the mean ± SD. Whenever necessary, the statistical significance of the data was analyzed by performing one-sample $t$ tests. The specific types of tests and $P$-values, when applicable, are indicated in the figure legends.

### Image processing

For immunostaining, the digital images obtained by the SD-OSR microscope were processed using Photoshop 7.0 (Adobe). The projections of z-stack images were processed with Metamorph. For time-lapse imaging, the digital images obtained by the SD-OSR microscope were processed and projected using ImageJ. Images were prepared from tiff-tagged files (8-bit grayscale or 24-bit RGB) using Photoshop 7.0.

## Supplementary Information

## Acknowledgements

We are grateful to our laboratory members for fruitful discussions. We especially thank M Uji, Y Sugiyama, and F Takenaga for technical assistance in our laboratory. We are grateful to Drs M Takeichi (Riken, BDR) and A Nagafuchi (Nara Medical University) for the generous gift of the rat anti-E-cadherin (ECCD2) mAb and rat anti-$\alpha$-catenin ($\alpha$18) mAb, respectively. We thank Dr T Uemura (Kyoto University) for discussions. This research was supported by Grants-in-Aid for Scientific Research (A) (JP18H03999) to S Tsukita, for Scientific Research (B) (JP16H05121) to A Tamura, for Young Scientists (B) (JP17K17853) to S Konishi, for Young Scientists (B) (JP18K14696) to T Yano, for Japan Society for the Promotion of Science Research Fellow (JP18J22965) to H Kanoh, from Japan Society for the Promotion of Science, Takeda Science Foundation and Uehara Memorial Foundation to S Tsukita, and Core Research for Evolutional Science and Technology (JP115811) from the Japan Science and Technology Agency (JST) to S Tsukita.

## Author Contributions

S Konishi: conceptualization, data curation, formal analysis, investigation, methodology, and project administration.
T Yano: conceptualization, data curation, investigation, formal analysis, methodology, and project administration.
H Tanaka: conceptualization, formal analysis, investigation, and methodology.
T Mizuno: investigation.
H Kanoh: formal analysis.
K Tsukita: investigation and methodology.
T Namba: data curation, formal analysis, investigation, and methodology.
A Tamura: conceptualization, formal analysis, investigation, and methodology.
S Yonemura: data curation, formal analysis, and investigation.
S Gotoh: investigation.
H Matsumoto: investigation.
T Hirai: investigation.
S Tsukita: conceptualization, formal analysis, funding acquisition, investigation, methodology, and project administration.

## Conflict of Interest Statement

The authors declare that they have no conflict of interest.

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
