## [Reviewer comments · Life Science Alliance]

Life Science Alliance

Vinculin is critical for the robustness of the epithelial cell sheet paracellular barrier for ions

Satoshi Konishi, Tomoki Yano, Hiroo Tanaka, Tomoaki Mizuno, Hatsuho Kanoh, Kazuto Tsukita, Toshinori Namba, Atsushi Tamura, Yonemura Shigenobu, Shimpei Gotoh, Hisako Matsumoto, Toyohiro Hirai, and Sachiko Tsukita

DOI: <https://doi.org/10.26508/lsa.201900414>

Corresponding author(s): Sachiko Tsukita, Osaka University

Review Timeline:	Submission Date:	2019-05-04
	Editorial Decision:	2019-05-04
	Revision Received:	2019-07-21
	Editorial Decision:	2019-07-23
	Revision Received:	2019-07-23
	Accepted:	2019-07-24

Scientific Editor: Andrea Leibfried

Transaction Report:

Please note that the manuscript was previously reviewed at another journal and the reports were taken into account in the decision-making process at Life Science Alliance.

Reviewer #1 Review

Comments to the Authors (Required):

The manuscript by Konishi and colleagues shows that loss of Vinculin by KO of the gene in Eph4 monolayers results in a loss of barrier to diffusion of ions, likely by causing junctional instability.

Whereas there is very little to comment on the technical aspects of this study, one may wonder (as this reviewer does) whether this observed effect of vinculin depletion really increases our understanding of vinculin function or of epithelial barrier function. This wonder is increased by the experiments performed to clarify the reason for the loss of ion-barrier upon disruption of the vinculin gene.

Basically, these experiments show that actomyosin activity, through pulling on junctional complexes induces a tension-induced protective feedback function of vinculin, which is needed for junction stability. There is a significant body of work including multiple reviews that showed and discussed exactly that. It is very strange to note that none of this work is properly cited in this manuscript. Some newness could be in a specific regulation of the apical junctional complex by vinculin, but the experiments showing that overexpression of α -catenin or an α -catenin-vinculin fusion can restore the ion-barrier argue that this is in fact due to an underlying instability of the adherens junction complex. The suggestion that the apically located interaction between vinculin and ZO-1 is important is not by far conclusively experimentally tested in this study.

Taken together, this study may hint at a novel piece in the puzzle of junction regulation, but without substantial further molecular insight cannot be regarded as novel.

Reviewer #2 Review

Comments to the Authors (Required):

Konishi et al report a role for vinculin in regulating epithelial barrier permeability. Strikingly, they show that deleting vinculin selectively disrupts the barrier for ions (measured by TER), but not for macromolecules. They show that the linear morphology of both TJ and AJs is altered, although by EM (including freeze-fracture EM) the TJs do not seem grossly altered. Instead, they attribute the functional changes to a change in the distribution of mechanical forces throughout the junctions. Using the α -18 mAb as a proxy of molecular tension, they report that this redistributes to tricellular junctions in vinculin KO cells. Further, cell shape is unstable and inhibiting myosin restores the ionic barrier dysfunction to vinculin KO monolayers.

Overall, this adds to recent reports that suggest that mechanical forces influence barrier function. But, while the data are lovely, I think that the MS is too preliminary at the moment for this journal.

Major comments

1. The characterization of mechanical patterns is not yet strong enough to be convincing.
 - a) The α -18 Ab is a useful proxy, but it needs to be corrected for changes in α -catenin distribution (i.e. co-stained and normalized for α -catenin). (i.e. The authors need to exclude the possibility that α -18 is changing because α -catenin itself is being redistributed.) Furthermore, the data needs to be presented quantitatively.
 - b) Patterns of tension need to be corroborated with other measures of tension, such as the use of FRET-based tension sensors in AJ proteins (for which a number are now available).
2. It would be useful to also consider whether the relevant properties that are being altered are the rigidity or stiffness of the junctions. Tension in actin cables can make them more rigid. I could imagine that if junctions were softer or less rigid, then they might display the fluctuations that the authors capture in their movies.
3. They do not have clear evidence for a mechanism by which vinculin would regulate the distribution of mechanical force at junctions.
 - a) It is interesting that the barrier dysfunction can be partially rescued by targeting the D5 region of vinculin (which bears an actin-binding capacity) to AJ by fusion with α -catenin. This is a good start, but doesn't itself give clear insight into what might be happening. Does it alter F-actin at junctions

and what parameter(s) of the junctional F-actin are being changed (amount, stability, dynamics, organization). And do these transgenes correct the abnormal patterns of tension (whatever they may be)? These are the kind of questions that could provide useful insights.

4. Where is vinculin acting to have these effects? The authors propose that "Vinculin at AJs may interact with the via ZO-1" (p 8, para 1). This suggestion is based on the observation that exogenously expressed vinculin and ZO-1 can interact in HEK293 cells (Fig S2C). Although an interesting start, it really is not enough to support even a mild suggestion. The analysis was not done in epithelial cells; there was no evidence to suggest that the vinculin also interacted with AJ proteins; and ZO-1 has been reported to associate with AJ proteins, such as a-catenin, so it is not clear that this represents TJ-associated ZO-1.

5. Where is the barrier defect occurring? Is it at the tricellular junctions or in the bicellular areas between the vertices? Perhaps the new labelling assay that Ann Miller's lab recently published in Dev Cell could help here.

Small points

"Mechanical fluctuations". I find the authors' use of this term to be rather ambiguous. Sometimes (e.g. para 2 of Introduction) they seem to use it to mean change in mechanical forces over time (the first meaning that would occur to me). But in the results (e.g. p9, para 1) it seems to mean spatial changes (here for a-18 Ab staining).

Fig 2B. The EM data seem poorly contrasted to my eyes, at least in the files submitted.

Fig 4C seems to be missing a caption.

Reviewer #3 Review

Comments to the Authors (Required):

Konishi et al describe a phenotypic characterization of vinculin-KO mammary epithelial cells (Eph4). They show a role for vinculin in the regulation of tight junction (TJ) paracellular permeability to ions, as measured by TER, but not to larger solutes, as measured by fluorescent dextran assays. They find that in vinculin-KO cells the apical junctional complex stochastically shows dynamically distorted patterns. How this translates into altered permeability to ions, however, remains unclear.

The role of the actomyosin cytoskeleton and of AJ in driving the assembly and regulation of the TJ barrier has been shown by a number of studies. Thus, the concept that "TJ and AJ act collaboratively in the apical junctional complex" (abstr.) is not new. However, how precisely TJ membrane components integrate inputs from TJ-associated versus adherens junctions (AJ)-dependent actomyosin contractility, and how TJ components affect actomyosin tension at AJ and vice-versa are important questions, that remain to be investigated. This manuscript addresses the function of an AJ protein, vinculin, which is associated with belt-like AJ. Previous work by the Takeichi, Niessen and Miller laboratories highlighted the role of vinculin as a protein that accumulates at the apical junctional complex when and where (cytokinesis, upper layers of the epidermis) TJ are formed, and/or are under dynamic mechanical stress. The observation, reported in this manuscript that cells lacking vinculin have disorganized bicellular and especially tricellular junctions, and that this correlates with increased permeability to ions, but not to larger molecules is

interesting. However, it remains largely phenomenological and descriptive, and does not provide a sufficiently deep mechanistic insight into how vinculin contributes to regulating TJ barrier function.

Major issues

- Vinculin has been reported to interact with ZO-1 (see also scheme in Fig. 5D). Yet, no experiment in this paper addresses, through rescue, mutation, dynamic or other experiments if and how the interaction of vinculin with ZO-1 is implicated in the phenotype. Only immunofluorescence localization is provided. However, distribution patterns of proteins alone are not sufficient to establish mechanisms.
- Vinculin is localized at AJ, and its absence could potentially affect the function not only of ZO-1, but also alpha-catenin and afadin, that are ZO-1-interacting proteins. In fact, no mechanistic link is provided to ZO proteins or any additional TJ protein (cytoplasmic or membrane), besides immunofluorescence of a limited number of markers.
- The movies highlight the observation that tricellular junctions are particularly affected by the KO of vinculin. Could the increased passage of ions be due to tTJ, rather than bTJ-dependent permeability?
- The immune-EM observation, reported here, that ZO-1 is localized both at AJ and at TJ in confluent epithelial cells raises additional questions: does ZO-1 change its localization (by immune-EM) upon KO of vinculin? In page 9 the authors state that vinculin mechanically fine-tunes the "position" of the AJC, but no experiment is carried out to validate this assertion.
- The phenotype can be rescued by overexpressing alpha-catenin. This raises issues of redundancy, that must be addressed.
- Does vinculin act to reduce junctional tension, and if so, can this be demonstrated experimentally (eg. TsMod cadherin sensor)? Just concluding that vinculin "fine-tunes" or "resists" the actomyosin-dependent fluctuations does not provide significant advance in understanding mechanistically how it is implicated in mechanical transduction to TJ.
- The authors say that the TER values of vinculin-KO cells (400-500 Ohms) are "very close" to the value of dZO-KO cells, but this is difficult to accept, in view of the fact that only one clonal line of KO cells is described here. Characterization of additional clonal lines must be provided, to make sure that no clonal-dependent effects are observed.
- Was a microarray analysis performed on these cells, and could there be increased expression of one or more specific claudin isoforms that could account for altered permeability to ions?

Other issues

- Results, Page 6. The authors state that TJ strands imaged by freeze fracture are "more or less formed" in the WT and vinculin-KO cells. What does "more or less" mean? Can the authors provide quantitative data? Also, just below, they say that the distorted linear TJ pattern were in "the limited areas"? Any quantification possible?
- ZO1 is used, but the correct spelling is ZO-1.
- The authors state that stress fibers were not affected in vinculin-KO cells, but in the image shown they appear slightly increased (Fig. 2C). Some form of quantification would help here as well.
- The alpha-18 antigen staining appears increased especially at tTJ in vinculin-KO cells (with respect to WT) Figure 4A, but decreased in Figure 4C.
- Figure S1B the x-axis lacks the label.

May 4, 2019

Re: Life Science Alliance manuscript #LSA-2019-00414-T

Prof. Sachiko Tsukita
Osaka University
Graduate School of Frontier Biosciences and Graduate School of Medicine
Yamadaoka 2-2
Suita 565-0871
Japan

Dear Dr. Tsukita,

Thank you for transferring your manuscript entitled "Vinculin is critical for the robustness of the epithelial cell sheet paracellular barrier for ions" to Life Science Alliance. The manuscript was assessed by expert reviewers at another journal before, and the editors transferred these reports to us with your permission.

The reviewers appreciated the quality of your work, but thought that the mechanistic insight provided remains rather limited. This concern does not preclude publication in Life Science Alliance, and we would thus like to invite you to submit a revised version to us. We would expect a point-by-point response to all concerns raised and accordingly text changes/further discussion and alteration of data representation as well as addressing experimentally point 1a of reviewer #2 and bullet points 4, 7 and the request for quantifications in 'Other issues' of reviewer #3.

Thank you for this interesting contribution to Life Science Alliance. We are looking forward to receiving your revised manuscript.

Sincerely,

Andrea Leibfried, PhD
Executive Editor
Life Science Alliance
Meyerhofstr. 1
69117 Heidelberg, Germany
t +49 6221 8891 502

B. MANUSCRIPT ORGANIZATION AND FORMATTING:

Manuscript

"Vinculin is critical for the robustness of the epithelial cell sheet paracellular barrier for ions "

Responses to the Reviewers' Comments

We appreciate the constructive comments given by the editors and reviewers, and have done our best to incorporate the suggested changes as thoroughly as possible. In particular, we have added a substantial amount of new data to address the editor's and reviewers' concerns. We believe that the manuscript has been greatly improved.

Reviewer #1

The manuscript by Konishi and colleagues shows that loss of Vinculin by KO of the gene in Eph4 monolayers results in a loss of barrier to diffusion of ions, likely by causing junctional instability. Whereas there is very little to comment on the technical aspects of this study, one may wonder (as this reviewer does) whether this observed effect of vinculin depletion really increases our understanding of vinculin function or of epithelial barrier function. This wonder is increased by the experiments performed to clarify the reason for the loss of ion-barrier upon disruption of the vinculin gene. Basically, these experiments show that actomyosin activity, through pulling on junctional complexes induces a tension-induced protective feedback function of vinculin, which is needed for junction stability. There is a significant body of work including multiple reviews that showed and discussed exactly that. It is very strange to note that none of this work is properly cited in this manuscript.

We appreciate the constructive comments given by the reviewer. We revised the manuscript to correct it according to the comment and added the references and relevant information from them (Page 3; Second paragraph of the revised manuscript).

Major point

Point 1. Some newness could be in a specific regulation of the apical junctional complex by vinculin, but the experiments showing that overexpression of α -catenin or an α -catenin-vinculin fusion can restore the ion-barrier argue that this is in fact due to an underlying instability of the adherens junction complex. The suggestion that the apically located interaction between vinculin and ZO-1 is important is not by far conclusively experimentally tested in this study. Taken together, this study may hint at a novel piece in the puzzle of junction regulation, but without substantial further molecular insight cannot be regarded as novel.

We are grateful for this comment. We consider it is very important to investigate the interaction between vinculin and ZO-1. We showed the result of the co-immunoprecipitation assay of vinculin and ZO-1 in HEK cells in the submitted manuscript (Fig S2C, Page 7; Last paragraph ~ Page8; First paragraph of the revised manuscript). Furthermore, the results of immuno-EM show that vinculin and ZO-1 are localized at the apical junctional complex. Thus these results suggest the interaction between ZO-1 and vinculin.

Reviewer #2

Konishi et al report a role for vinculin in regulating epithelial barrier permeability. Strikingly, they show that deleting vinculin selectively disrupts the barrier for ions (measured by TER), but not for macromolecules. They show that the linear morphology of both TJ and AJs is altered, although by EM (including freeze-fracture EM) the TJs do not seem grossly altered. Instead, they attribute the functional changes to a change in the distribution of mechanical forces throughout the junctions. Using the α -18 mAb as a proxy of molecular tension, they report that this redistributes to tricellular junctions in vinculin KO cells. Further, cell shape is unstable and inhibiting myosin restores the ionic barrier dysfunction to vinculin KO monolayers. Overall, this adds to recent reports that suggest that mechanical forces influence barrier function. But, while the data are lovely, I think that the MS is too preliminary at the moment for this journal.

Major points

Point 1. The characterization of mechanical patterns is not yet strong enough to be convincing.
 a) The α -18 Ab is a useful proxy, but it needs to be corrected for changes in α -catenin distribution (i.e. co-stained and normalized for α -catenin). (i.e. The authors need to exclude the possibility that α -18 is changing because α -catenin itself is being redistributed.) Furthermore, the data needs to be presented quantitatively.

We appreciate these important observations. We added our immunohistochemistry results for α -18 co-stained and normalized to α -catenin, and added the quantitative data to the figures and text (Figure 4A and B)(Page 9:Second paragraph of the revised manuscript). We could rule out the possibility that α -18 changed because of the redistribution of α -catenin itself. We confirmed that the signal intensity of α -18 is not significantly different between a bicellular point and a tricellular point in wildtype Eph4 cells, and that the signal intensity of α -18 is higher at a tricellular point than at a bicellular point in vinculin KO Eph4 cells.

b) Patterns of tension need to be corroborated with other measures of tension, such as the use of FRET-based tension sensors in AJ proteins (for which a number are now available).

As the reviewer affirms, it is important to know the patterns of tension depending on the state of the Eph4 cells. It is difficult for us to perform such experiments to analyze the junctional tension now. Nevertheless, we take it as a pending issue to be solved in the future.

Point 2. It would be useful to also consider whether the relevant properties that are being altered are the rigidity or stiffness of the junctions. Tension in actin cables can make them more rigid. I could imagine that if junctions were softer or less rigid, then they might display the fluctuations that the authors capture in their movies.

We do agree with the reviewer and think that measuring the rigidity or stiffness of the junction is important. Thus, this issue should be discussed carefully. In general, it is technically difficult to carry out that kind of experiments by now, but we will not lose this point and will consider it in future investigations.

Point 3. They do not have clear evidence for a mechanism by which vinculin would regulate the distribution of mechanical force at junctions. a) It is interesting that the barrier dysfunction can be partially rescued by targeting the D5 region of vinculin (which bears an actin-binding capacity) to AJ by fusion with α -catenin. This is a good start, but doesn't itself give clear insight into what might be happening. Does it alter F-actin at junctions and what parameter(s) of the junctional F-actin are being changed (amount, stability, dynamics, organization). And do these transgenes correct the abnormal patterns of tension (whatever they may be)? These are the kind of questions that could provide useful insights.

We see the relevance in investigating the change of the parameters of F-actin (amount, stability, dynamics and organization). Unfortunately, for us it is technically difficult to clearly investigate them now. We plan to address this issue in the future.

Point 4. Where is vinculin acting to have these effects? The authors propose that "Vinculin at AJs may interact with the via ZO-1" (p 8, para 1). This suggestion is based on the observation that exogenously expressed vinculin and ZO-1 can interact in HEK293 cells (Fig S2C). Although an interesting start, it really is not enough to support even a mild suggestion. The analysis was not done in epithelial cells; there was no evidence to suggest that the vinculin also interacted with AJ proteins; and ZO-1 has been reported to associate with AJ proteins, such as α -catenin, so it is not clear that this represents TJ-associated ZO-1.

We thank for valuable comment. In the current work, the interaction of exogenously expressed ZO1 and vinculin was confirmed in HEK cells. This result could indicate the interaction of vinculin and ZO1 in epithelial cells, possibly at TJs and AJs, because vinculin and ZO-1 are localized and concentrated in the apical junctional complex. However, it is not a result obtained using epithelial cells. Following this, we changed the description about the interaction of ZO-1 and vinculin in the manuscript (Page 7; Last paragraph ~ Page8; First paragraph).

Point 5. Where is the barrier defect occurring? Is it at the tricellular junctions or in the bicellular areas between the vertices? Perhaps the new labelling assay that Ann Miller's lab recently published in Dev Cell could help here.

We appreciate this comment which raise a key and very interesting issue. By now, it is technically difficult for us to find out where exactly the barrier defect is occurring. To analyze and quantify the difference of the barrier defect both in the bicellular and the tricellular point is on our list for future investigations.

Minor points

Point 6. "Mechanical fluctuations". I find the authors' use of this term to be rather ambiguous. Sometimes (e.g. para 2 of Introduction) they seem to use it to mean change in mechanical forces over time (the first meaning that would occur to me). But in the results (e.g. p9, para 1) it seems to mean spatial changes (here for α -18 Ab staining).

As pointed out by the reviewer, we use the term "Mechanical fluctuations" to mean change in mechanical forces. We can visualize mechanical forces generated by actomyosin using the anti- α -catenin monoclonal antibody (α 18), which binds to mechanically stretched α -catenin (Yonemura et al., 2010). Thus, the α -18 stainings also showed that change in mechanical forces.

Point 7. Fig 2B. The EM data seem poorly contrasted to my eyes, at least in the files submitted.

We readjusted the contrast for those images, especially the ones of E-cadherin and Cingulin (Fig 2B).

Point 8. Fig 4C seems to be missing a caption.

We added the caption of Fig 4C (page 29 of the revised manuscript).

Reviewer #3

Konishi et al describe a phenotypic characterization of vinculin-KO mammary epithelial cells (Eph4). They show a role for vinculin in the regulation of tight junction (TJ) paracellular permeability to ions, as measured by TER, but not to larger solutes, as measured by fluorescent dextran assays. They find that in vinculin-KO cells the apical junctional complex stochastically shows dynamically distorted patterns. How this translates into altered permeability to ions, however, remains unclear.

The role of the actomyosin cytoskeleton and of AJ in driving the assembly and regulation of the TJ barrier has been shown by a number of studies. Thus, the concept that "TJ and AJ act collaboratively in the apical junctional complex" (abstr.) is not new. However, how precisely TJ membrane components integrate inputs from TJ-associated versus adherens junctions (AJ)-dependent actomyosin contractility, and how TJ components affect actomyosin tension at AJ and vice-versa are important questions, that remain to be investigated. This manuscript addresses the function of an AJ protein, vinculin, which is associated with belt-like AJ. Previous work by the Takeichi, Niessen and Miller laboratories highlighted the role of vinculin as a protein that accumulates at the apical junctional complex when and where (cytokinesis, upper layers of the epidermis) TJ are formed, and/or are under dynamic mechanical stress. The observation, reported in this manuscript that cells lacking vinculin have disorganized bicellular and especially tricellular junctions, and that this correlates with increased permeability to ions, but not to larger molecules is interesting. However, it remains largely phenomenological and descriptive, and does not provide a sufficiently deep mechanistic insight into how vinculin contributes to regulating TJ barrier function.

Major points

Point 1 and 2. Vinculin has been reported to interact with ZO-1 (see also scheme in Fig. 5D). Yet, no experiment in this paper addresses, through rescue, mutation, dynamic or other experiments if and how the interaction of vinculin with ZO-1 is implicated in the phenotype. Only immunofluorescence localization is provided. However, distribution patterns of proteins alone are not sufficient to establish mechanisms.

Vinculin is localized at AJ, and its absence could potentially affect the function not only of ZO-1, but also alpha-catenin and afadin, that are ZO-1-interacting proteins. In fact, no mechanistic link is provided to ZO proteins or any additional TJ protein (cytoplasmic or membrane), besides immunofluorescence of a limited number of markers.

We are grateful for this comment. We consider it is very important to investigate the interaction between vinculin and ZO-1. We showed the result of the co-

immunoprecipitation assay of vinculin and ZO-1 in HEK cells in the submitted manuscript (Fig S2C, Page 7; Last paragraph ~ Page8; First paragraph of the revised manuscript). Furthermore, the results of immuno-EM show that vinculin and ZO-1 are localized at the apical junctional complex. Thus these results suggest the interaction between ZO-1 and vinculin.

Point 3. The movies highlight the observation that tricellular junctions are particularly affected by the KO of vinculin. Could the increased passage of ions be due to tTJ, rather than bTJ-dependent permeability?

We appreciate this noteworthy comment. It is generally difficult in a technical level to carry out experiments to investigate the increased passage of ions difference between bTJ and tTJ. However, it is a crucial question to answer. Therefore, we are willing to assess this in the future.

Point 4. The immune-EM observation, reported here, that ZO-1 is localized both at AJ and at TJ in confluent epithelial cells raises additional questions: does ZO-1 change its localization (by immune-EM) upon KO of vinculin? In page 9 the authors state that vinculin mechanically fine-tunes the "position" of the AJC, but no experiment is carried out to validate this assertion.

We performed an experiment of immunoelectron microscopy of ZO-1 in vinculin KO Eph4 cells, and added the image as supplementary information (Fig. S2D). We also compared this result to the one obtained using wildtype Eph4 and found that there was no significant difference in the distribution of ZO-1 protein. Given this, we changed the linked content of the text (Page 8; First paragraph).

Point 5. The phenotype can be rescued by overexpressing alpha-catenin. This raises issues of redundancy, that must be addressed.

We absolutely agree with the idea proposed by the reviewer. We think the result of the rescue of phenotype by overexpressing alpha-catenin in vinculin KO Eph4 cells may be because of the similarity of the molecular structure of alpha-catenin to that of vinculin. However, we are considering it for future research projects. Hence, we did not show this point in the present manuscript.

Point 6. Does vinculin act to reduce junctional tension, and if so, can this be demonstrated experimentally (eg. TsMod cadherin sensor)? Just concluding that vinculin "fine-tunes" or "resists" the actomyosin-dependent fluctuations does not provide significant advance in understanding mechanistically how it is implicated in mechanical transduction to TJ.

Thank you for constructive comment. It is technically difficult to investigate the junctional tension at the moment, so we will take it into account in next studies.

Point 7. The authors say that the TER values of vinculin-KO cells (400-500 Ohms) are "very close" to the value of dZO-KO cells, but this is difficult to accept, in view of the fact that only one clonal line of KO cells is described here. Characterization of additional clonal lines must be provided, to make sure that no clonal-dependent effects are observed.

We generated 6 different vinculin KO Eph4 cells by using 3 different guide RNAs, as we described in the materials and methods section. We could confirm significant reduction of the TER values of all vinculin KO cells. We emphasized the description about this issue in the main manuscript (page 5, Second paragraph of the revised manuscript). We also could see the difference of the TER values of vinculin KO cell sheets and the TJ-less ZO1/ZO2 DKO cell sheets. We added the description of the difference to the revised manuscript (page 5, line 31-33).

Point 8. Was a microarray analysis performed on these cells, and could there be increased expression of one or more specific claudin isoforms that could account for altered permeability to ions?

We compared the expression of mRNA of all the 27 claudin family subtypes between WT Eph4 cells and vinculin KO Eph4 cells by quantitative RT-PCR. We could not find the decrease of the expression of barrier type claudin and could not see the significant increase of channel type claudin including claudin-2, -10, -15, -21 in vinculin KO Eph4 cells compared to WT Eph4 cells. So, we think that the change of expression of claudin is not the cause of altered permeability of vinculin KO Eph4 cells.

Minor points

Point 9. Results, Page 6. The authors state that TJ strands imaged by freeze fracture are "more or less formed" in the WT and vinculin-KO cells. What does "more or less" mean? Can the authors provide quantitative data?

We conducted the quantitative analysis of TJ strands both in wildtype and vinculin KO Eph4 cells by measuring junctional depth and strands number, according to previous reports (Coyner et al., 2002). There were no significant differences in junctional depth and strands number between wildtype and vinculin KO cells. Thus, we added the result of the experiment to the figure and revised manuscript carefully to address this observation (Page 6, Second paragraph).

Point 10. Also, just below, they say that the distorted linear TJ pattern were in "the limited areas"? Any quantification possible?

We quantified and compared the number of distorted TJ areas between wildtype and vinculin KO Eph4 cell sheets (Fig. 1D).

Point 11. ZO1 is used, but the correct spelling is ZO-1.

Thank you for noting this. We changed the description of ZO1 to ZO-1 in the manuscript.

Point 12. The authors state that stress fibers were not affected in vinculin-KO cells, but in the image shown they appear slightly increased (Fig. 2C). Some form of quantification would help here as well.

We quantified the number of stress fiber and compared the number of wildtype Eph4 cell and Vinculin KO Eph4 cells. There was no significant difference (Fig. 2D).

Point 13. The alpha-18 antigen staining appears increased especially at tTJ in vinculin-KO cells (with respect to WT) Figure 4A, but decreased in Figure 4C.

We quantitatively analyzed the α -18 intensity at the bicellular and tricellular TJs, and suggested that α -18 intensity was increased especially at tricellular TJ in vinculin-KO cells. In addition, we changed the images in order to show the phenotype more clearly (Fig. 4D).

Point 14. Figure S1B the x-axis lacks the label.

We labeled the days on the x-axis of Fig. S1B.

July 23, 2019

RE: Life Science Alliance Manuscript #LSA-2019-00414-TR

Prof. Sachiko Tsukita
Osaka University
Medical School
Yamadaoka 2-2, Suita, Osaka 565-0871
Osaka 565-0871
Japan

Dear Dr. Tsukita,

Thank you for submitting your revised manuscript entitled "Vinculin is critical for the robustness of the epithelial cell sheet paracellular barrier for ions". I have now evaluated the revised version of your manuscript. I appreciate your response to the reviewer concerns and the additional analyses performed and would be happy to publish your paper in Life Science Alliance pending final revisions necessary to meet our formatting guidelines.

- please add callouts to figure S3C in the manuscript text
- please upload all figure files as individual files

A. FINAL FILES:

-- Summary blurb (enter in submission system): A short text summarizing in a single sentence the study (max. 200 characters including spaces). This text is used in conjunction with the titles of papers, hence should be informative and complementary to the title. It should describe the context and significance of the findings for a general readership; it should be written in the present tense

and refer to the work in the third person. Author names should not be mentioned.

B. MANUSCRIPT ORGANIZATION AND FORMATTING:

Sincerely,

July 24, 2019

RE: Life Science Alliance Manuscript #LSA-2019-00414-TRR

Prof. Sachiko Tsukita
Osaka University
Medical School
Yamadaoka 2-2, Suita, Osaka 565-0871
Osaka 565-0871
Japan

Dear Dr. Tsukita,

Thank you for submitting your Research Article entitled "Vinculin is critical for the robustness of the epithelial cell sheet paracellular barrier for ions". It is a pleasure to let you know that your manuscript is now accepted for publication in Life Science Alliance. Congratulations on this interesting work.

DISTRIBUTION OF MATERIALS:

Again, congratulations on a very nice paper. I hope you found the review process to be constructive and are pleased with how the manuscript was handled editorially. We look forward to future exciting submissions from your lab.

Sincerely,
